# Tilt Angle Adjustment for Incident Solar Energy Increase: A Case Study for Europe

**Dragos Machidon * and Marcel Istrate**

Power Engineering Department, Faculty of Electrical Engineering, "Gheorghe Asachi" Technical University, 700050 Iasi, Romania
* Correspondence: machidon.dragos@tuiasi.ro

**Abstract:** The work presented in this paper aims to analyze the efficiency of using optimum tilt angles defined for several time intervals to maximize the incident solar irradiation on a certain surface deployed in Europe. Such a technique would improve the solar energy harvesting process, both for photovoltaic panels and solar thermal collectors, while not investing in the more expensive sun-tracking devices. The optimum tilt angles have been determined for the yearly, bi-annual, seasonal, and monthly time intervals, which were delimited on a calendar and astronomical base, respectively, considering multiple locations from Europe, and using various mathematical models based on empirical equations and solar irradiation estimation. The total incident solar irradiation provided by adjusting the tilt angle multiple times per year was calculated and compared to that obtained when using the yearly optimum tilt angle. The gains for each type of adjustment were investigated, and the monthly optimization of the tilt angle is obviously the most effective one, ensuring gains of up to 7%, depending on the considered latitude. Otherwise, an optimization twice per year, using the bi-annual optimum tilt angles determined for the astronomical-defined warm and cold seasons, would be the next best solution.

**Keywords:** renewable energy; solar irradiation; optimum tilt angle; sustainable development





## 1. Introduction

### 1.1. Background

The quest for a more proper usage of the solar potential is crucial for the worldwide efforts to sustain the growing energy demand [1] and to mitigate the pressing climate changes [2], thus ensuring sustainable development of the entire world. It is obvious that for the aforementioned goals to be achieved, a significant increase in terms of installed photovoltaic (PV) panels is required [3], but, at the same time, several solutions must be found to increase the PV panels' energy output as well, especially considering that the ongoing climate changes are reducing the solar potential, according to various reports all over the world [4–7].

When it comes to a better harvesting the solar energy, the proper orientation of a PV panel or a solar thermal collector towards the sun is the first condition that must be fulfilled for ensuring the highest energy output. For instance, a PV panel will best perform when the solar irradiation enters it at an angle of 90°, or very close to it, but for this to happen sun-tracking systems must be used. They will indeed provide the proper orientation of the PV panels in azimuth and elevation angles, thus ensuring a notable increase in the energy output, of 12 to 25% for the single-axis tracking systems [8], and 30 to 45% for the dual-axis tracking systems, according to [9–11]. Nonetheless, they also come with significant costs with the initial investment and with the maintenance later [12]. Due to these drawbacks, the sun-tracking systems are not used on a large scale, as in 2019 there were only 25.25 GW of installed power in PV panels using such tracking devices (22.05 GW on single-axis systems and 3.2 GW on double-axis systems) [13], which represents only 4.31% of the 584.68 GW installed worldwide in the same year [14].

On the other hand, installing the PV panels at fixed angles toward the ground is not the best solution, but it is significantly cheaper than using tracking devices. Usually, the PV panels and solar collectors are installed at a fixed, yearly optimum tilt angle ($\beta_y$), towards the ground, which is considered to provide the best performances over the entire year. However, using $\beta_y$ will not ensure the highest amount of incident solar irradiation on the considered surface throughout the various seasons across the year. Consequently, one possible solution to increase the total amount of incident solar irradiation would be the possibility to manually adjust the tilt angle of the surface, considering other optimum tilt angles, such as bi-annual optimum tilt angles ($\beta_b$), seasonal optimum tilt angles ($\beta_s$) or even monthly optimum tilt angles ($\beta_m$).

The estimation of the optimum tilt angles for smaller periods of time was also of great interest among the scientific community, as various studies were conducted for different locations worldwide. Thus, in [15], a 6.6% increase in the annual energy output was observed after adjusting the PV panels' tilt according to $\beta_s$ determined for Lahore, Pakistan. In [16], the authors have determined that adjusting the PV panels' tilt angle according to $\beta_b$, $\beta_s$, and $\beta_m$ will increase the annual incident radiation with 10.5%, 10.7% and 11.7%. An 8% increase in incident radiation was determined when adjusting the tilt angle two times per year in Zahedan, Iran [17], and twelve times per year (monthly) in Madinah, Saudi Arabia [18]. In [19], the electricity output of a 1 MW sample PV plant was evaluated for various locations in Turkey, considering the manual adjustment of the PV panels' tilt angle according to the bi-annual, seasonal, and monthly optimum tilt angles, and the increase in the energy output was 3.21–3.71% (for $\beta_b$ angles), 3.64–4.26% (for $\beta_s$ angles), and of 4.53–5.3% (for $\beta_m$ angles), respectively. Although these gains may seem insignificant, the authors have determined that the monthly adjustment of the tilt angle will increase the internal return rate by 0.7–0.9% and reduce the discounted payback period by 8 to 10 months [20]. A monthly optimization of the PV panels' tilt angles using the levelized cost of energy (LCOE) criteria was performed in [21] for the cities of Tripoli (Lebanon), Belfort (France), and Tantan (Morocco), and it was found that the LCOE has decreased by 4.32%, 3.73%, and 4.35% when a monthly adjustment of the PV panels' tilt angle was performed in those specific locations.

The authors of this paper have conducted their own research determining the yearly, bi-annual, seasonal, and monthly optimum tilt angles for the city of Iasi, Romania, according to four different methods, and a nearly 5% increase in annual incident radiation and PV output was determined when considering using the monthly optimum tilt angles [22,23].

### 1.2. Objectives and Paper Structure

The main goal of this work is to investigate the efficiency of using various optimum tilt angles for increasing the solar irradiation on a tilted surface, deployed on multiple locations in Europe, between 35° to 60° latitude, and not only for a single specific location, as most of the current existing studies have investigated. At the same time, we intend to conduct a comparative analysis regarding the performances of several of the existing mathematical models from the literature, considering several simple empirical models and a more complex model based on the estimation of the solar irradiation, which will be defined as a solar irradiation search-based method (SBM). Additionally, when determining the $\beta_b$ and $\beta_s$ angles, two different scenarios for the bi-annual and seasonal time intervals will be considered, calendar-based and astronomical-based, respectively.

This technique could be applied to small size, or even medium size, photovoltaic systems, or solar collector systems, avoiding the significant costs required by the autonomous tracking systems in terms of both initial investment and further maintenance costs.

The paper is organized as follows: In Section 2 the mathematical models considered for the estimation of the yearly, bi-annual, seasonal, and monthly optimum tilt angles will be presented. In Section 3 the obtained values of the optimum tilt angles will be presented and discussed, and their performances will be evaluated in terms of the gains of the incident solar irradiation. In-depth comparative analyses will be performed to evaluate

the performances of these optimum angles for the considered latitudes. Finally, conclusions will be presented in Section 4.

## 2. Materials and Methods

The work presented in this paper aims to present a comparative analysis regarding the efficiency of using optimum tilt angles for shorter time intervals, such as two times per year (using bi-annual optimum tilt angles—$\beta_b$), four times per year (using seasonal optimum tilt angles—$\beta_s$), and twelve times per year (using monthly optimum tilt angles—$\beta_m$) in terms of total solar irradiation incident on a surface. To determine these various optimum tilt angles, several mathematical models from the literature were selected; the mathematical expressions of each model are presented in Table 1.

**Table 1.** Mathematical models for evaluating different optimum tilt angles.

| Optimum Angle | Mathematical Expression | | | | Author |
|---|---|---|---|---|---|
| **Yearly—$\beta_y$** | $\beta_y = \phi$ | | | | Patko [23] |
| | $\beta_y = -0.007209 \cdot \phi^2 + 1.096 \cdot \phi + 2.373$ | | | | C. Martin [24] |
| | $0.83\,\phi + 0.62$ | | | | Modarressi [25] |
| | $H_t = H_b \cdot R_b + H_d \cdot R_d + \rho H \cdot R_r$ | | | | SBM–Liu & Jordan [26] |
| **Bi-annual—$\beta_b$** | $H_t = H_b \cdot R_b + H_d \cdot R_d + \rho H \cdot R_r$ | | | | SBM–Liu & Jordan [26] |
| **Seasonal—$\beta_s$** | $\beta_{spring} = \beta_{autumn} = \phi$ <br> $\beta_{summer} = \phi - 23.45,\ \beta_{winter} = \phi + 23.45$ | | | | Patko [23] |
| | $\beta_{spring} = 0.80\,\phi - 15.67$ <br> $\beta_{summer} = 0.79\,\phi - 21.99$ <br> $\beta_{autumn} = 0.86\,\phi + 16.78$ <br> $\beta_{winter} = 0.87\,\phi + 23.46$ | | | | Modarressi [25] |
| | $H_t = H_b \cdot R_b + H_d \cdot R_d + \rho H \cdot R_r$ | | | | SBM–Liu & Jordan [26] |
| **Monthly—$\beta_m$** | $\beta_{m1-3} = 60.00012 + 1.49986\,M - 3.49996\,M^2 + (\phi - 30)(0.7901 + 0.01749\,M + 0.0165\,M^2)$ | | | | El-Kassaby [27] |
| | $\beta_{m4-6} = 216.0786 - 72.03219\,M + 6.00312\,M^2 + (\phi - 40)(1.07515 + 0.11244\,M - 0.03749\,M^2)$ | | | | |
| | $\beta_{m7-9} = 29.11831 - 20.52981\,M + 2.50186\,M^2 + (\phi - 50)(-11.17256 + 2.70569\,M - 0.15035\,M^2)$ | | | | |
| | $\beta_{m10-12} = -441.2385 + 84.54322\,M - 3.50196\,M^2 + (\phi - 40)(4.2137 - 054834\,M + 0.0223\,M^2)$ | | | | |
| | January | $0.88\,\phi + 27.61$ | July | $0.78\,\phi - 26.7$ | Modarressi [25] |
| | February | $0.86\,\phi + 17.88$ | August | $0.80\,\phi - 16.66$ | |
| | March | $0.84\,\phi + 3.83$ | September | $0.82\,\phi - 2.08$ | |
| | April | $0.81\,\phi - 11.52$ | October | $0.85\,\phi + 13.23$ | |
| | May | $0.78\,\phi - 23.61$ | November | $0.88\,\phi + 25.14$ | |
| | June | $0.77\,\phi - 29.15$ | December | $0.89\,\phi + 30.45$ | |
| | $H_t = H_b \cdot R_b + H_d \cdot R_d + \rho H \cdot R_r$ | | | | SBM–Liu&Jordan [26] |

The empirical model proposed by Patko in [23] was chosen due to its simplicity; this being the simplest possible way for evaluating the yearly optimum tilt angle and the seasonal optimum tilt angles for solar collectors. Thus, in the northern hemisphere, the optimum yearly tilt angle should be equal to the latitude, as well as for the optimum spring ($\beta_{spring}$) and autumn angles ($\beta_{autumn}$), while for the summer and winter optimum angles ($\beta_{summer}$, $\beta_{winter}$), the maximum value of the Earth's declination ($\delta = 23.45$) should be subtracted or added from the considered location latitude.

In Ref. [24], C. Martin proposed various empirical expressions as functions of latitude, based on polynomial regression, for evaluating the yearly optimum tilt angles worldwide. This model was considered in our analysis for its simplicity, worldwide application (the authors have considered irradiation data from almost 14,500 sites from all over the world [26]), and its robustness.

The model proposed by Modarresi in [25] was considered because the author has proposed simple empirical equations, not only for the estimation of the yearly optimum tilt angle, but also for the seasonal, and monthly optimum tilt angles, which are applicable to the northern hemisphere. In this sense, this model suits this paper's objectives very well.

El-Kassaby was one of the first researchers to propose mathematical expressions for calculating the values of the monthly optimum tilt angle in the late '80s, and his model [27] was chosen to be evaluated with other new models that have been developed since then. In his expressions, $M$ is the specific month, each equation in Table 1 being applicable for a set of three months (January–March, April–June, July–September, and October–December).

The search-based method (SBM) is in fact an iterative algorithm that will compute and compare the total incident radiation on a surface, while its tilt angle is varied between $0°$ and $90°$, with a certain step, usually of $1°$. This methodology is well known and used by many researchers in the field [24,28–30]. The general structure of the algorithm consists of the following steps:

- Read input data: time, location (latitude and longitude), daily total radiation on a horizontal surface ($H$), and constant values ($G_{sc}$, $\rho$);
- Set the surface's tilt angle ($\beta = 0°$);
- Compute the Earth's declination angle ($\delta$) and solar geometry ($\omega_s$, $\omega'_s$);
- Determine the solar irradiation components (beam, diffuse, and reflected) incident to the surface;
- Compute and store the value of the total incident radiation to the surface ($H_t$);
- Increase the tilt angle with $1°$ and run steps $3 \div 5$ again until $\beta = 90°$;
- Compare the values of the $H_t$ and select $\beta_{opt}$ as the angle for which $H_t$ has the highest value.

This procedure can be implemented for estimating the optimum tilt angle for any period, such as daily, monthly, seasonally, bi-annual, or yearly.

In this study, the total irradiation incident on a tilted surface for the considered period will be determined as the sum of the individual components, namely direct ($H_b$), diffuse ($H_d$), and reflected ($H_r$), considering the equation proposed by Liu and Jordan in [26], and presented in Table 1. In this model, an important parameter is the ratio between the average values of the diffuse component of the irradiation and the daily total irradiation on a horizontal ($H_d/H$) surface, which depends on the clearness index ($K$). Several expressions were proposed for this correlation, but for this study, the authors opted for the multi-location model proposed for Europe by Bortolini in [31], which is described by Equation (1).

$$\frac{\overline{H_d}}{\overline{H}} = 0.9888 - 0.395 \cdot \overline{K} - 3.7003 \cdot \overline{K}^2 + 2.2905 \cdot \overline{K}^3 \tag{1}$$

The efficiency of the various bi-annual, seasonal, and monthly optimum tilt angles, regarding a yearly tilt angle, will be evaluated considering the criterion of the annual maximum incident irradiation received by a surface when tilted according to the optimum tilt angles previously determined.

The methods previously described will be used to determine the optimum tilt angles of a south-facing surface, deployed in Europe, at various latitudes between $35°$ and $60°$, with a step of $5°$. When using the solar irradiation search-based method, the values of the daily total radiation on a horizontal surface ($H$) were obtained from the PVGIS–SARAH database [32].

## 3. Results and Discussions

In the first part of this section, the values of the optimum tilt angles will be presented, and after that, their use efficiency will be analyzed.

### 3.1. Optimum Tilt Angles for Europe

Considering the models presented in the previous section of the paper, several algorithms were developed to calculate the optimum tilt angles. The obtained results are further presented.

a. Yearly optimum tilt angles ($\beta_y$)

Four models have been considered for the $\beta_y$ estimation: the empirical model proposed by Patko (PTK) [23], the regression model proposed by C. Martin (CM) [24], the regression model proposed by Modarresi (MOD) [25] and the radiation search-based method (SBM). The yearly optimum tilt angles obtained are presented in Table 2.

**Table 2.** Yearly optimum tilt angles.

| Model | Latitude (°) | | | | | |
|---|---|---|---|---|---|---|
| | **35** | **40** | **45** | **50** | **55** | **60** |
| **PTK** | 35 | 40 | 45 | 50 | 55 | 60 |
| **CM** | 31.9 | 34.7 | 37.1 | 39.2 | 40.9 | 42.2 |
| **MOD** | 29.7 | 33.8 | 38 | 42.1 | 46.3 | 50.4 |
| **SBM** | 31 | 31 | 34 | 37 | 39 | 45 |

As the model proposed by Patko assumes that the optimum tilt angle is equal to the latitude, it will provide the highest values of $\beta_y$, of all four models. A quite good correlation can be observed in values provided by the model proposed by C. Martin and the search-based method.

b. Bi-annual optimum tilt angles ($\beta_b$)

The bi-annual optimum tilt angles have been determined using only the radiation search-based method (SBM), considering two distinct seasons along the year, namely the warm season and the cold season. However, these two seasons have been delimited on the following assumptions:

- Calendar-based: the *warm season* extends from 1 March to 30 September, while the *cold season* extends from 1 October to 28 February.
- Astronomical definition: the *warm season* extends between the spring and autumn equinox (from 20 March to 22 September), while the *cold season* extends from 23 September to 19 March.

Thus, the SBM algorithm was tuned according to the mentioned assumptions, and the values obtained considering the calendar delimited seasons were noted as cSBM (calendar search-based method), while the other ones as aSBM (astronomical search-based method). The values of the bi-annual optimal tilt angles are presented in Table 3.

**Table 3.** Values of the bi-annual optimum tilt angles.

| Season | Model | Latitude (°) | | | | | |
|---|---|---|---|---|---|---|---|
| | | **35** | **40** | **45** | **50** | **55** | **60** |
| Warm | cSBM | 10 | 13 | 17 | 20 | 23 | 26 |
| | aSBM | 11 | 16 | 20 | 24 | 28 | 33 |
| Cold | cSBM | 56 | 59 | 64 | 67 | 72 | 78 |
| | aSBM | 55 | 57 | 62 | 65 | 69 | 76 |

As expected, the optimum tilt angles for the warm season are lower than those for the cold season. However, one can notice that when astronomical seasons are considered, the optimum tilt angles are slightly higher than those determined for the calendar-based seasons, for both warm and cold seasons.

c.   Seasonal optimum tilt angles ($\beta_s$)

A seasonal adjustment of the PV panel's tilt angle assumes that the year is split into four seasons, typically spring, summer, autumn, and winter, and the optimum tilt angle ($\beta_s$) is determined for each season.

In our analysis, we determined the value of $\beta_s$ for each season using the mathematical expressions previously presented in Table 1.

Similar to the bi-annual tilt angles, previously determined, when using the SBM algorithm, the seasons were delimited on a calendar-based assumption and according to the astronomical definition, as follows:

- Calendar-based: *Spring*—1 March to 30 May; *Summer*—1 June to 31 August; *Autumn*—1 September to 30 November; *Winter*—1 December to 28 February;
- Astronomical definition: *Spring*—20 March to 20 June, *Summer*—21 June to 22 September; *Autumn*—23 September to 20 December; *Winter*—21 December to 19 March.

The calculated values of $\beta_s$ are presented in Table 4 for the considered latitudes.

**Table 4.** Seasonal optimum tilt angles for latitudes in Europe.

| Season | Model | Latitude (°) | | | | | |
|--------|-------|------|------|------|------|------|------|
| | | 35 | 40 | 45 | 50 | 55 | 60 |
| Spring | PTK | 35.0 | 40.0 | 45.0 | 50.0 | 55.0 | 60.0 |
| | MOD | 12.3 | 16.3 | 20.3 | 24.3 | 28.3 | 32.3 |
| | cSBM | 19.0 | 21.0 | 25.0 | 28.0 | 31.0 | 33.0 |
| | aSBM | 8.0 | 11.0 | 15.0 | 18.0 | 21.0 | 22.0 |
| Summer | PTK | 11.5 | 16.5 | 21.5 | 26.5 | 31.5 | 36.5 |
| | MOD | 5.7 | 9.6 | 13.6 | 17.5 | 21.5 | 25.4 |
| | cSBM | 5.0 | 10.0 | 16.0 | 20.0 | 25.0 | 30.0 |
| | aSBM | 14.0 | 19.0 | 26.0 | 31.0 | 36.0 | 42.0 |
| Autumn | PTK | 35.0 | 40.0 | 45.0 | 50.0 | 55.0 | 60.0 |
| | MOD | 46.9 | 51.2 | 55.5 | 59.8 | 64.1 | 68.4 |
| | cSBM | 51.0 | 54.0 | 60.0 | 64.0 | 70.0 | 77.0 |
| | aSBM | 57.0 | 60.0 | 65.0 | 68.0 | 73.0 | 80.0 |
| Winter | PTK | 58.5 | 63.5 | 68.5 | 73.5 | 78.5 | 83.5 |
| | MOD | 53.9 | 58.3 | 62.6 | 67.0 | 71.3 | 75.7 |
| | cSBM | 58.0 | 60.0 | 63.0 | 66.0 | 69.0 | 74.0 |
| | aSBM | 54.0 | 55.0 | 59.0 | 61.0 | 64.0 | 69.0 |

Of all models, the one proposed by Patko will provide the highest values for the $\beta_s$ in spring, and the lowest values in autumn. Changing how seasons are delimited has a significant impact on the optimum tilt values. As one can notice in Table 4, when using the astronomical definition $\beta_s$, there will be lower values in winter and spring, and higher values in summer and autumn, compared to those obtained under the calendar-based scenario.

d.   Monthly optimum tilt angles ($\beta_m$)

The monthly optimum tilt angles have also been determined according to the three models presented in Table 1, and the obtained values of $\beta_m$ are presented in Table 5 for latitudes between 35° and 45°, and in Table 6 for the other latitudes.

**Table 5.** Monthly optimum tilt angles for latitudes between 35° and 45°.

| Month | Latitude (°) | | | | | | | | |
|---|---|---|---|---|---|---|---|---|---|
| | **35** | | | **40** | | | **45** | | |
| | KSB | MOD | SBM | KSB | MOD | SBM | KSB | MOD | SBM |
| January | 62.1 | 58.4 | 61 | 66.2 | 62.8 | 63 | 70.4 | 67.2 | 67 |
| February | 53.5 | 48.0 | 52 | 57.9 | 52.3 | 55 | 62.4 | 56.6 | 59 |
| March | 38.0 | 33.2 | 38 | 42.9 | 37.4 | 41 | 47.9 | 41.6 | 46 |
| April | 19.4 | 16.8 | 19 | 24 | 20.9 | 23 | 28.6 | 24.9 | 28 |
| May | 2.5 | 3.7 | 3 | 6 | 7.6 | 8 | 9.5 | 11.5 | 12 |
| June | 0 | 0 | 0 | 0 | 1.7 | 0 | 2 | 5.5 | 2 |
| July | 2 | 0.6 | 0 | 4 | 4.5 | 3 | 6 | 8.4 | 8 |
| August | 12.2 | 11.3 | 13 | 16.5 | 15.3 | 18 | 20.7 | 19.3 | 23 |
| September | 32.0 | 26.6 | 31 | 37 | 30.7 | 35 | 42 | 34.8 | 41 |
| October | 49.2 | 43.0 | 48 | 54 | 47.2 | 51 | 58.8 | 51.5 | 56 |
| November | 60.6 | 55.9 | 59 | 65 | 60.3 | 61 | 69.4 | 64.7 | 65 |
| December | 64.8 | 61.6 | 63 | 69 | 66.1 | 65 | 73.2 | 70.5 | 69 |

**Table 6.** Monthly optimum tilt angles for latitudes between 50° and 60°.

| Month | Latitude (°) | | | | | | | | |
|---|---|---|---|---|---|---|---|---|---|
| | **50** | | | **55** | | | **60** | | |
| | KSB | MOD | SBM | KSB | MOD | SBM | KSB | MOD | SBM |
| January | 74.5 | 71.6 | 69 | 78.6 | 76 | 73 | 82.7 | 80.4 | 79 |
| February | 66.8 | 60.9 | 62 | 71.3 | 65.2 | 66 | 75.7 | 69.5 | 73 |
| March | 52.8 | 45.8 | 50 | 57.8 | 50 | 54 | 62.7 | 54.2 | 61 |
| April | 33.3 | 29.0 | 32 | 37.9 | 33 | 37 | 42.5 | 37.1 | 41 |
| May | 13 | 15.4 | 16 | 16.5 | 19.3 | 20 | 20 | 23.2 | 24 |
| June | 4 | 9.4 | 6 | 6 | 13.2 | 10 | 8 | 17.1 | 14 |
| July | 8 | 12.3 | 11 | 10 | 16.2 | 15 | 12 | 20.1 | 18 |
| August | 25 | 23.3 | 26 | 29.3 | 27.3 | 30 | 33.5 | 31.3 | 33 |
| September | 47 | 38.9 | 44 | 52 | 43 | 49 | 57 | 47.1 | 54 |
| October | 63.6 | 55.7 | 60 | 68.4 | 60 | 65 | 73.2 | 64.2 | 70 |
| November | 73.8 | 69.1 | 68 | 78.2 | 73.5 | 73 | 82.6 | 77.9 | 79 |
| December | 77.4 | 75 | 72 | 81.7 | 79.4 | 75 | 85.9 | 83.9 | 81 |

The model proposed by El-Kassaby (KSB) and the search-based model (SBM) provides the closest results as their mean bias error (MBE) is −1.34°, while MBE between SBM and MOD is 1.45°, and between KSB and MOD, it is 2.79°. The model proposed by El-Kassabi tends to provide higher optimum tilt angles from January to April and from August to December for most of the latitudes; the maximum difference being under 7° for 60° latitude, and smaller values at latitudes over 40° for the summer months.

### 3.2. Efficiency of Using Bi-Annual, Seasonal, and Monthly Optimum Tilt Angles

While some comparisons between the angles determined according to various models have been presented in the previous section, the main objective of this study is to evaluate the efficiency of several adjustments (two, four, and twelve) of a surface tilt angle

throughout the year. This analysis was conducted by determining the annual total solar irradiation received by a surface that is tilted according to the bi-annual, seasonal, and monthly optimum tilt angles (previously presented in Tables 3–6) and comparing it with the annual total irradiation provided by the yearly optimum tilt angle (from Table 2).

For the determination of solar irradiation, a new algorithm was developed, considering the Liu and Jordan mathematical model. The results obtained when using the yearly optimum tilt angles and the bi-annual tilt angles are presented in Table 7, and the results obtained when seasonal and monthly optimum tilt angles have been considered are presented in Table 8.

**Table 7.** Annual total solar irradiation on a surface tilted at yearly and bi-annual optimum angles.

| Lat. (°) | $H_t$ (kWh/m$^2$) | | | | | |
|---|---|---|---|---|---|---|
| | Yearly Optimum Angles | | | | Bi-Annual Angles | |
| | PTK | CM | MOD | SBM | cSBM | aSBM |
| 35 | 2186.44 | 2190.52 | 2190.45 | 2190.79 | 2302.63 | 2320.92 |
| 40 | 1765.49 | 1779.80 | 1781.07 | 1783.19 | 1858.03 | 1871.08 |
| 45 | 1666.12 | 1686.01 | 1684.95 | 1687.43 | 1747.52 | 1763.53 |
| 50 | 1317.05 | 1341.76 | 1338.35 | 1342.65 | 1382.49 | 1395.53 |
| 55 | 1165.98 | 1197.42 | 1191.35 | 1197.78 | 1228.83 | 1242.14 |
| 60 | 1076.49 | 1103.97 | 1100.60 | 1104.63 | 1130.65 | 1149.15 |

**Table 8.** Annual total solar irradiation on a surface tilted at seasonal and monthly optimum angles.

| Lat. (°) | $H_t$ (kWh/m$^2$) | | | | | | |
|---|---|---|---|---|---|---|---|
| | Seasonal Angles | | | | Monthly Angles | | |
| | PTK | MOD | cSBM | aSBM | KSB | MOD | SBM |
| 35 | 2291.47 | 2314.15 | 2318.21 | 2319.33 | 2347.50 | 2345.40 | 2348.01 |
| 40 | 1846.50 | 1868.01 | 1868.76 | 1869.56 | 1890.84 | 1890.67 | 1891.98 |
| 45 | 1740.10 | 1758.84 | 1758.21 | 1759.80 | 1782.51 | 1781.63 | 1783.68 |
| 50 | 1372.88 | 1391.60 | 1390.50 | 1391.23 | 1409.10 | 1409.27 | 1410.68 |
| 55 | 1217.63 | 1238.24 | 1235.39 | 1238.97 | 1253.77 | 1254.12 | 1255.44 |
| 60 | 1125.91 | 1142.66 | 1137.19 | 1141.12 | 1162.10 | 1161.24 | 1163.55 |

When the yearly optimum tilt angles are compared, it can be observed that those provided by the SBM algorithm are ensuring the highest values of the total incident irradiation, and the absolute differences between the use of SBM angles and those predicted by the other three models are presented in Figure 1.

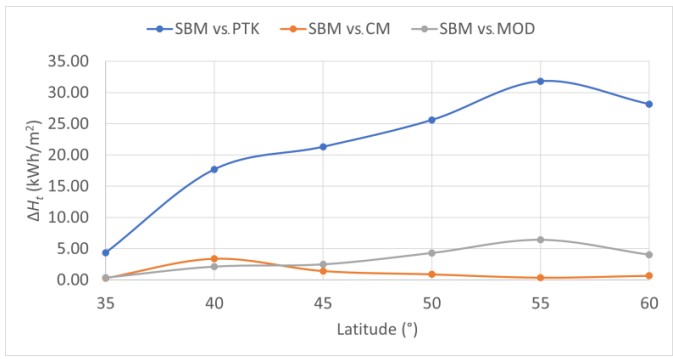

**Figure 1.** Differences in incident irradiation between the use of SBM angles and those determined by PTK, CM, and MOD models.

As one can notice, the differences between using the angles proposed by the considered methods are under 10 kWh/m², apart from the model proposed by Patko, which underperforms at latitudes higher than 40°, as the differences in terms of incident irradiation grow to over 15 kWh/m², the maximum difference being over 32 kWh/m² at 55° latitude. A very good correlation can be observed between the results obtained using the angle determined by SBM and the regression model proposed by C. Martin.

Similar conclusions can be drawn when considering the data in Table 8, as the model proposed by Patko ensures, once again, the smallest values of the incident radiation (as it was the case when the yearly optimum tilt angles have been analyzed—see Figure 1). However, the differences between the results provided by the other two models are rather insignificant. Hence, one can conclude that these simple models, based on empirical equations, could be successfully used to determine the optimum tilt angles for any time interval wanted and any location, thus avoiding the complexity of the radiation search-based methods.

The impact of multiple changes of tilt angle across the year has been analyzed considering the optimum tilt angles determined according to the SBM algorithm. The annual total irradiation of the yearly optimum tilt angle was considered as a reference, and the increase in incident irradiation on the surface, when using bi-annual, seasonal and monthly tilt angles has been determined, both in absolute and relative values. Thus, in Table 9, the absolute gains of incident radiation by all latitudes are presented, while Figure 2 depicts these gains in relative values.

**Table 9.** Gains of incident irradiation due to tilt angle's optimization.

| Angle | $\Delta H_t$ (kWh/m²) | | | | | |
|---|---|---|---|---|---|---|
| | Latitude (°) | | | | | |
| | 35 | 40 | 45 | 50 | 55 | 60 |
| $\beta_{bc}$ | 111.8 | 74.8 | 60.1 | 39.8 | 31.1 | 26.0 |
| $\beta_{ba}$ | 130.1 | 87.9 | 76.1 | 52.9 | 44.4 | 44.5 |
| $\beta_{sc}$ | 127.4 | 85.6 | 70.8 | 47.8 | 37.6 | 32.6 |
| $\beta_{sa}$ | 128.5 | 86.4 | 72.4 | 48.6 | 41.2 | 36.5 |
| $\beta_{m}$ | 157.2 | 108.8 | 96.2 | 68.0 | 57.7 | 58.9 |

$\beta_{bc}$, $\beta_{ba}$—bi-annual optimum tilt angle for calendar-based seasons and astronomical-based seasons. $\beta_{sc}$, $\beta_{sa}$—seasonal optimum tilt angle for calendar-based seasons and astronomical-based seasons. $\beta_{m}$—monthly optimum tilt angle.

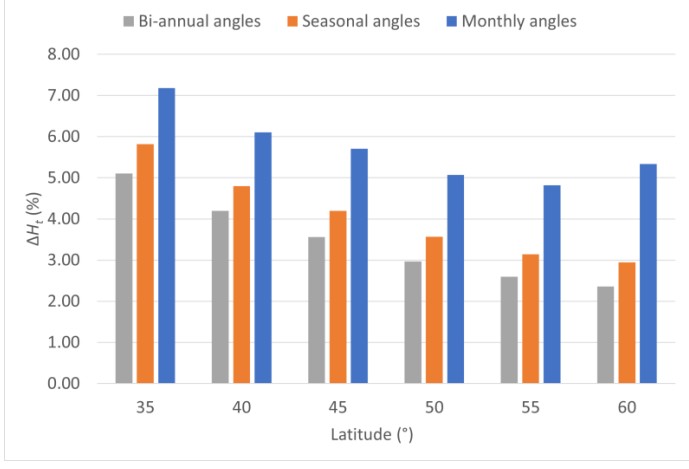

**Figure 2.** Relative incident irradiation gains for bi-annual, seasonal, and monthly tilt angle optimization.

One can notice that the monthly optimization of the tilt angles ensures the best performances, as expected, but its efficiency is not the same for all latitudes across Europe, as the highest increases of total incident radiation (over 100 kWh/m$^2$) are reported at 35° and 40° latitude, with a maximum of 157.2 kWh/m$^2$ at 35°, while at lower and higher latitudes, the gains of incident radiation are less important.

A similar pattern can be observed for the bi-annual and seasonal angles as well, as such an optimization of the surface's tilt angle will boost the incident irradiation, especially at lower latitudes (35° and 40°). In relative values, the increase in the incident solar irradiation due to the monthly optimization of the tilt angle is slightly over 7% at 35° latitude, with values lower than 6% at the other analyzed latitudes.

At the same time the efficiency of seasonal optimum angles is rather marginal when compared to that of the bi-annual angles, the gains of solar irradiation being under 20 kWh/m$^2$, at 35° latitude and even smaller in the rest.

### 3.3. Analysis of the Impact of How the Seasons Are Defined

Another important aspect that must be discussed is the impact of how the seasons are defined on the total incident radiation. Back when the seasonal and bi-annual optimum tilt angles were determined, we considered two different scenarios: calendar-based and astronomical. When analyzing the data in Table 9, one can observe that the use of the optimum tilt angles determined according to the astronomical definition of the seasons proves to be more efficient. In fact, the use of bi-annual optimum tilt angles determined for the astronomically defined seasons ($\beta_{ba}$) will ensure a higher total incident radiation even when the calendar-based seasonal optimum tilt angles ($\beta_{sc}$) are used, as shown in Figure 3.

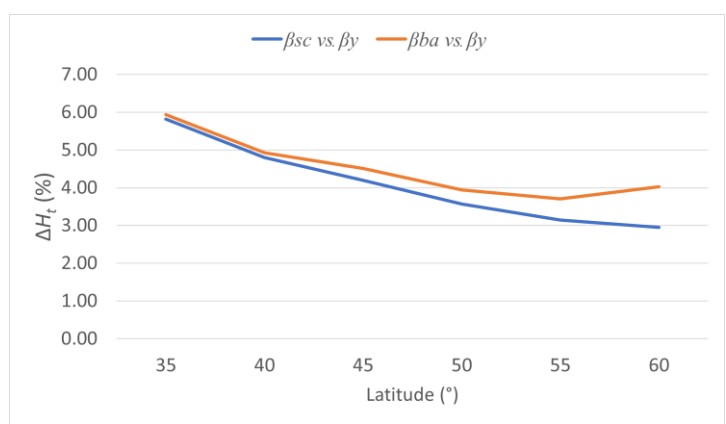

**Figure 3.** Comparison between the relative increase in incident radiation when $\beta_{ba}$ and $\beta_{sc}$ angles are used.

This is a very important observation, which strongly supports the idea that two manual adjustments of the PV panel's tilt angle along the year, according to the warm and cold astronomical-defined seasons, are even more efficient than adjusting the tilt angle four times per year.

### 4. Conclusions

An investigation regarding the efficiency of periodically adjusting the tilt angle of a surface exposed to the sun was conducted over multiple latitudes across Europe, considering a 5° step in latitude. The yearly, bi-annual, seasonal, and monthly optimum tilt angles have been determined according to various mathematical models proposed over time by several researchers. The efficiency of manually changing the surface's tilt angle multiple times per year was analyzed based on the increase in the annual total incident irradiation, when compared to the total irradiation provided by the yearly optimum tilt angle.

The yearly optimum tilt angles for each latitude across Europe could be easily determined with the empirical equation proposed by C. Martin in [26], as they will ensure

similar performances to the angles determined with the more complex radiation search-based method.

However, for best performances, we recommend the monthly optimization of the tilt angle, for each latitude across Europe, as it will ensure the highest values for the annual total incident irradiation. The monthly optimum tilt angles should be determined using the radiation search-based method (SBM) for the most accurate values, although the empirical models proposed by El-Kassaby and Modarresi are far easier to use, and their results are very close to those provided by the SBM method.

If the monthly adjustment of the PV panels' (or solar collectors) tilt angle may seem like a chore for some of the owners or administrators of such systems, based on this study, we can recommend an optimization two times per year, using the bi-annual optimum tilt angles determined for the astronomical-defined warm (from 20 March to 22 September) and cold seasons (from 23 September to 19 March). In this way, the complexity of the PV panels' mounting system and the effort put in for adjusting the tilt angles are minimal, and the incident radiation gain is only slightly reduced when compared to the use of monthly tilt angles.

The obtained results could prove very useful because this technique can be easily implemented for the small size PV and the solar thermal systems in residential areas, as the owners could adjust the tilt angle themselves several times per year. Positive energy communities could emerge in this way, as people will play an active role by ensuring better harvesting of solar energy and will better understand the notion of direct involvement in the transition towards a low-carbon power system. On a larger scale, even medium size PV systems could implement this technique, as it would take only minutes for the operating staff personnel to manually adjust the tilt angle.

**Author Contributions:** Conceptualization, D.M. and M.I.; methodology, D.M.; software, D.M.; validation, D.M. and M.I.; formal analysis, M.I.; investigation, D.M.; resources, D.M.; data curation, D.M. and M.I; writing—original draft preparation, D.M. and M.I.; writing—review and editing, D.M.; visualization, M.I.; supervision, D.M.; project administration, D.M. All authors have read and agreed to the published version of the manuscript.

**Funding:** This research received no external funding.

**Institutional Review Board Statement:** Not applicable.

**Informed Consent Statement:** Not applicable.

**Data Availability Statement:** The data regarding the values of the daily total radiation on a horizontal surface were obtained from the PVGIS—SARAH database.

**Acknowledgments:** Special thanks to all who contributed to writing the manuscript and the reviewers suggestions for improving the manuscript's quality.

**Conflicts of Interest:** The authors declare no conflict of interest.

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
