# Peer review of "Tilt Angle Adjustment for Incident Solar Energy Increase: A Case Study for Europe"

_sustainability, doi:10.3390/su15087015_

Round 1

Reviewer 1 Report

I would like to congratulate the authors on taking up an interesting topic that may influence higher energy yields from photovoltaic installations. On the one hand, the information is known, but collecting many models and showing the optimal values of inclination angles for different latitudes is the basis for PV system owners. Maybe there is no "great scientific news" in this publication, but the information is important and useful.

In the literature review, you can also include various publications related to the topic of the work, e.g.:

https://doi.org/10.3390/en8021025

https://doi.org/10.3390/en14102938

https://doi.org/10.3390/en14164964

https://doi.org/10.1016/j.renene.2020.02.010

Author Response

Dear esteemed Reviewer

Thank you very much for the time and dedication you put into reviewing our work. We appreciate your suggestions, as they will help us improve the current paper’s manuscript, and future other papers as well. All the modifications made have been marked red in the manuscript's revised version, which is attached to this response.

Reviewer 2 Report

1. The methodology of Liu and Jordan section 2.1 is widely used in research and can therefore simply be presented by reference.

2. The considered models for determining the optimal tilt angle for various periods of time would be better tabulated (lines 224-261) for a simpler visual analysis.

3. Also, the results may be presented in the form of surfaces, where along the x and y axes plot the latitude and longitude of the locality, and along the z axis - the intensity of solar radiation for the period under consideration.

4. Also in the Conclusion section, it is necessary to indicate the areas of applicability of each model for determining the optimal tilt angle of the photovoltaic module, and what values are recommended for use for the territory under consideration.

5. Lines 280-286 should be moved to the methodology section

6. The methods under consideration are only briefly mentioned in the Introduction section (Line 64, 98). From the point of view of the authors, the considered methods for determining the optimal tilt angle should have a number of advantages, since the authors have chosen them. But nothing is said about this in the article.

Author Response

(The authors gave the same response as above.)

Reviewer 3 Report

The subject matter of the article is of interest to the journal, but needs to be significantly modified for publication. The introduction is too long and does not provide enough information, so it should be modified and significantly reduced. The method has an excessive mathematical burden, the equations should be correctly referenced in the text and those that are not completely necessary should be eliminated in order to lighten the text and promote its comprehension. The results and discussion section is very long, its structure should be modified and summarised considerably, and graphs that do not provide relevant information should be eliminated. The figures and graphs should be modified to establish figures of higher quality and of a size in accordance with the text. The conclusions should be rewritten, as they are very brief and do not reflect the objectives achieved in the work carried out. In addition, the references should be properly revised so that they are all in the correct format.

Author Response

(The authors gave the same response as above.)

Reviewer 4 Report

The manuscript presents modelling studies on the determination of optimal tilt angles for photovoltaic and solar applications. The Authors studied and compared a variety of models showing the differences in irradiation gains using them. Data on irradiation were taken from a well-known database (PVGIS). The paper is well-written and organized. In my opinion, it can be interesting for researchers and investors in the field and thus it could be published in Sustainability.

Some detailed comments:

  • The main disadvantage, in my opinion, of the paper is the lack of presentation of any experimental (measured) data to verify some of the models (including the literature ones). Thus, line 285 'real measured data for all the considered locations were not available' is not really true. Indeed, many studies on PV performance or degradation rates include measurements of irradiance in the plane of area (tilted surface). It shouldn't be difficult to find such results and compare them with the results of the simulations. 
  • Unit (°) should be added to latitude values in figures.
  • I would avoid using the word 'below' in line 362 because you never know where finally your figure is going to be placed in the paper.
  • instead of using 'useful conclusions' (l. 18), it would be better to point out some of the most important

Author Response

(The authors gave the same response as above.)

Round 2

Reviewer 2 Report

All comments are addressed. The paper can be accept. 

Author Response

Thank you very much for your appreciation and acknowledgment.

Reviewer 3 Report

We are grateful for the modifications made to the paper, the quality of the submitted manuscript has been improved. A series of small changes are suggested for its publication, considering adapting the format appropriately, as there are inadequate spaces on page 10, as well as considering improving the figures to increase their quality.

Author Response

Thank you very much for your acknowledgment.  

We thoroughly checked our manuscript and found no inadequate spaces on page 10. However, we want to mention that our original manuscript was heavily edited by the journal's editor in charge of this submission and many alterations have occurred in this process (some of the tables are totally messed up in the edited manuscript, and blank tables have appeared, along with a lot of blank lines).   We will keep in touch with the journal's editor to address all these inconsistencies, although we believe they are part of the editing process.  

We also checked again and the dimensions and the resolution of the figures are consistent with the journal’s instruction guide regarding the figures’ format. The only problem seems to be when converting the Word manuscript into pdf format when indeed, the quality of the figures appears reduced. Also, we will keep in touch with the journal's editor to address this problem and find an adequate solution.
